# Prevention and management of anaemia in pregnancy: Community perceptions and facility readiness in Ghana and Uganda

Tara Tancred[1]*, Vincent Mubangizi[2], Emmanuel Nene Dei[3], Syliva Natukunda[4], Daniel Nana Yaw Abankwah[5], Phoebe Ellis[1], Imelda Bates[1], Bernard Natukunda[6], Lucy Asamoah Akuoko[3]

1 Department of International Public Health, Liverpool School of Tropical Medicine, Liverpool, United Kingdom, 2 Department of Family Medicine and Community Practice, Mbarara University of Science and Technology, Mbarara, Uganda, 3 Research, Planning, Monitoring & Evaluation Department, National Blood Service Ghana, Accra, Ghana, 4 Community Health Department, Mbarara University of Science and Technology, Mbarara, Uganda, 5 Department of Health Policy Planning and Management, University of Ghana School of Public Health, Accra, Ghana, 6 Department of Medical Laboratory Science, Mbarara University of Science and Technology, Mbarara, Uganda

* tara.tancred@lstmed.ac.uk

**Data Availability Statement:** All data can be found in the manuscript and supporting information files.

## Abstract

Anaemia is one of the most common conditions in low- and middle-income countries, with prevalence increasing during pregnancy. The highest burden is in Sub-Saharan Africa and South Asia, where the prevalence of anaemia in pregnancy is 41.7% and 40%, respectively. Anaemia in pregnancy can lead to complications such as prematurity, low birthweight, spontaneous abortion, and foetal death, as well as increasing the likelihood and severity of postpartum haemorrhage. Identifying and mitigating anaemia in pregnancy is a public health priority. Here we present a mixed-methods situational analysis of facility readiness and community understanding of anaemia in Ghana and Uganda. Quantitative health assessments (adapted from service availability and readiness assessments) and qualitative key informant interviews (KIIs) with district-level stakeholders, in-depth interviews (IDIs) with maternity staff, and focus group discussions (FGDs) with community members were held in 2021. We carried out facility assessments in nine facilities in Ghana and seven in Uganda. We carried out seven KIIs, 23 IDIs, and eight FGDs in Ghana and nine, 17, and five, respectively, in Uganda. Many good practices and general awareness of anaemia in pregnancy were identified. In terms of bottlenecks, there was broad consistency across both countries. In health facilities, there were gaps in the availability of haemoglobin testing—especially point-of-care testing—staffing numbers, availability of standard operating procedures/guidelines for anaemia in pregnancy, and poor staff attitudes during antenatal care. Amongst community members, there was a need for improved sensitisation around malaria and helminth infections as potential causes of anaemia and provision of education around the purpose of iron and folic acid supplementation for preventing or managing anaemia in pregnancy. Anaemia in pregnancy is a persistent challenge, but one with clear opportunities to intervene to yield improvements.

**Funding:** This work was wholly funded by the Medical Research Council, UK (reference MR/T00326X/1) as part of the Public Health Intervention Development scheme, awarded to IB. The funders had no role in study design, data collection and analysis, decision to publish, or preparation of the manuscript

**Competing interests:** The authors have declared that no competing interests exist.

## Introduction

Anaemia occurs when there are insufficient red blood cells or haemoglobin to carry oxygen to the tissues of the body. The prevalence of anaemia in women of reproductive age has been slow to improve. A review found the pooled prevalence of anaemia across 24 low-resource countries amongst women aged 15–24 was 42% [1]. Anaemia in pregnant women—sometimes called maternal, obstetric, or prepartum anaemia—is defined as a haemoglobin less than 110 g/L. It is an indicator of both poor nutrition and poor health. Globally, 36.8% of pregnant women are anaemic. The highest burden is in Sub-Saharan Africa and South Asia, where the prevalence of anaemia in pregnancy is 41.7% and 40% respectively [2].

Anaemia in pregnancy is associated with adverse outcomes for the mother and baby. These include preterm delivery, low birthweight infants, and impaired child development [3]. Severe anaemia, defined as haemoglobin below 60 g/L, can further result in spontaneous abortions, puerperal sepsis, and foetal death [4,5]. A review of anaemia in pregnancy in low-resource settings found that 12% of low birthweight births, 19% of preterm births, and 18% of perinatal mortalities were attributable to anaemia in pregnancy [6]. Severe anaemia also presents a risk to the mother, increasing the odds of postpartum haemorrhage by approximately 3.5 times [7].

Anaemia has multiple causes including: nutritional deficiencies (iron, folate, vitamin A); parasitic infections (malaria, hookworm, schistosomiasis); bleeding; underlying chronic conditions (tuberculosis, human immunodeficiency virus); and haemoglobinopathies such as sickle-cell disease [8]. The most common cause of anaemia in pregnancy is iron deficiency anaemia, though this may be proportionally less prevalent in malaria-endemic areas [5]. Iron deficiency can be caused by inadequate iron-rich food in the diet, or by blood loss, including secondary blood loss to helminth infections [9,10]. Women infected with intestinal parasites are 3.59 times more likely than uninfected women to develop anaemia, and women who have no iron and folic-acid supplementation are 1.82 times more likely to develop anaemia than supplemented women. Anaemia tends to worsen as pregnancy progresses, such that women in their third trimester are 2.37 times more likely to develop anaemia than those in the first and second trimesters [5].

Anaemia in pregnancy is a complex public health challenge, though there are many points from community-to-facility where helpful interventions may be put in place [11]. During pregnancy, antenatal care (ANC) is the main platform through which anaemia is identified, prevented, and treated. During antenatal visits, anaemia is assessed clinically or through the measurement of haemoglobin. Iron and folate supplements are generally prescribed, and, where appropriate, antimalarial prophylaxis and anti-helminth medication may be advised. With good adherence, this approach can improve iron stores and prevent or resolve anaemia in pregnancy [12]. Counselling on nutrition and malaria prevention, and encouragement to eat iron-rich foods, are also provided. For severe anaemia unresponsive to oral treatment, iron infusions, and occasionally, blood transfusions, may be warranted [5]. To meaningfully prevent or manage anaemia in pregnancy, it is critically important that ANC is accessible and of high quality, uptake is consistent, and that adherence to prescriptions and counselling is followed.

To strengthen health systems to catalyse reductions in anaemia in pregnancy, it is crucial to understand both facilitators and bottlenecks of anaemia care from multiple perspectives. With the aim of identifying good practices and areas for improvement, this study focused on perspectives about the prevention and management of anaemia in pregnancy amongst pregnant (or recently pregnant) women, male partners, community influencers, health services staff, and decision-makers who influence anaemia care policy and practice in Ghana and Uganda.

## Materials and methods

### Study design

We carried out a cross-sectional mixed methods situational analysis of factors from community-to-district that impact the prevention, management, and treatment of postpartum haemorrhage in Uganda and Ghana from March–May of 2021. The results here are derived from an embedded sub-study that focused on anaemia in pregnancy, given that it is a key and persistent factor influencing the likelihood and severity of postpartum haemorrhage.

### Study sites

Ghana and Uganda were selected as our study countries to maximize the diversity of contexts studied. As such, we wanted to select a country from West Africa and one from East Africa with different models of health system financing and governance. Due to longstanding research relationships in Ghana and Uganda across our study team, we selected these two countries. In both countries, we collected data from two districts. We piloted tools in an additional district in Ghana and an additional facility in Uganda. To facilitate the transferability of our findings, these districts were chosen to be "mid-range" in terms of population and geography compared to others in their respective countries. Among all the health facilities within each district—including private, public and faith-based—we selected only those that provided caesarean sections and blood transfusions, as we were interested in understanding readiness to support prevent, manage, and treat postpartum haemorrhage in referral-level facilities which are most likely to receive such cases. Our findings therefore reflect a census across all facilities in each study district meeting these criteria.

### Data collection

**Quantitative data collection.**   The study involved a quantitative health facility assessment based on adapted "health facility service availability and readiness assessments" [13]. This assessment had three different modules: one for anaemia, one for postpartum haemorrhage management, and one for blood transfusion—the one for anaemia is presented here. This module asked a comprehensive set of questions to determine the availability of appropriate standards or protocols, drugs, equipment, trained staff, and infrastructure to support anaemia prevention and management. Certain practices that are documented—or were expected to be documented—in patient files were also captured, for example, the measurement of haemoglobin at the time of labour. For such measures, we typically assessed all relevant patient files in the preceding three months. Within each health facility, different questions within the module were administered to maternal health care providers, laboratory technicians, or pharmacists as relevant, and where needed, observation (e.g. physically viewing a protocol) took place. No processes of care, however, were observed.

**Qualitative data collection.**   Qualitative data were derived from community focus group discussions (FGDs), in-depth interviews (IDIs), and key informant interviews (KIIs). FGDs were held in each district with: currently pregnant or recently delivered women; male partners of currently pregnant or recently delivered women; community leaders and elders (referred to hereafter as "community influencers"); and blood donors. FGD participants were recruited from communities within the catchment area of study sites in the health facility assessment. Study information was shared with community leaders, who then advertised it and helped identify prospective participants meeting study inclusion criteria in their communities to ensure 6–10 eligible persons would be available per focus group. Other than the provision of refreshments and transport reimbursements, participants were not unduly incentivised to

participate. FGDs explored perceptions about anaemia (how it is locally understood, what causes it, what its consequences are, how it can be prevented or managed) and ANC (why women do or do not attend, what occurs during ANC, its importance, and its role in anaemia prevention and management). IDIs were conducted with maternity in-charges and the heads of the participating health facilities to understand perceived facilitators and barriers to good practice in anaemia prevention and management. Laboratory technicians were also involved in IDIs to understand testing for anaemia and ordering blood for transfusion. There were typically only one or two people fulfilling each role in each participating facility, and we identified and interviewed them where possible. KIIs were carried out with key district- or national-level informants who play a role in supporting maternal and/or blood transfusion services in each participating district, including district health officers, district heads of maternity/reproductive health services, district heads of laboratory services, and regional or zonal blood transfusion leads. IDIs and KIIs were complementary to the health facility assessments, serving also to explain any gaps identified. All participants were purposively sampled based on their role in their community, health facility, or district and their lived experiences. Each participant participated in only one interview or group discussion.

FGDs took 60–90 minutes to complete, and IDIs and KIIs took 20–60 minutes to complete and were audio recorded. FGDs took place in convenient locations in each community where participants were derived from. To minimize the ask on participant time and travel, IDIs took place in the health facilities where the assessments were carried out with the relevant members of staff and KIIs were carried out in the offices of the key informants. All IDIs and KIIs were carried out in English, whilst FGDs were carried out in local languages as appropriate (Twi or Ga in Ghana and Runyankore in Uganda). All qualitative data were collected by skilled research assistants. They were registered nurses trained in maternal health, and in Ghana, they were also supported by staff from the National Blood Service for aspects pertaining to blood transfusion. They had at least three-years' of experience conducting health research within the study districts.

Our study team included clinicians specialising in obstetrics, haematology and blood transfusion, nurses, and social scientists, all with extensive experience in low- and middle-income country health services research. We developed our data collection instruments to explore some of the themes identified in the qualitative review of postpartum haemorrhage prevention by Finlayson et al [14] and the general processes described in the protocol by Akter et al [15]. We tailored them to the purpose and context of our study. Since tools needed minimal revision following piloting, we included data from the pilots in our analysis. Our data collection instruments can be found in "S1 Data Collection Instruments".

**Analysis.** Quantitative data were analysed in Excel to generate basic descriptive statistics (counts, averages, percentages). Qualitative data were read and re-read for familiarity. Framework analysis [16] was used to generate higher-level categories around specific pre-determined aspects of anaemia in pregnancy prevention and management. Data within each category were analysed thematically, being coded inductively, line-by-line in NVivo. These codes were grouped into increasingly higher-level codes to develop sub-themes to reflect key findings.

Quantitative data from the health facility assessments and qualitative data were then triangulated to present an overall picture of the key strengths and barriers to anaemia in pregnancy prevention and management across our study sites in Ghana and Uganda.

**Ethical considerations.** We obtained ethics approvals from the Ghana Health Service Ethics Review Committee and, in Uganda, from the Research Ethics Committee of Mbarara University of Science and Technology and the Uganda National Council for Science and Technology. In the UK, research ethics approval was obtained from the Liverpool School of Tropical Medicine.

Permission to carry out the research was obtained from district health offices and the head of each participating health facility. All participants provided written informed consent before proceeding with data collection.

## Results

Data were collected from a total of nine health facilities (inclusive of one pilot facility) in Ghana and seven (also inclusive of a pilot facility) in Uganda. As seen in Table 1, in Ghana, facilities from District 1 were generally larger than in other districts, with more patient traffic. Health facilities in Uganda had smaller overall patient numbers and tended to have facilities with larger capacity (as reflected by number of beds).

Qualitative data collection is summarised in Table 2 below. Data were collected from 84 participants in Ghana and 63 in Uganda.

### Diagnosing anaemia in pregnancy

Standard operating procedures or guidelines for assessing anaemia in pregnancy were not present in 7/9 (78%) facilities in Ghana, and in the other two facilities, respondents stated "unsure/don't know", or they were reported available but not seen. In Uganda, 5/7 (71%) facilities did not have these in place, and in the two other facilities, they were reported available but not seen (Table 3).

The health facility assessments reported various methods for testing women's haemoglobin concentration. The use of an automated haematology analyser for doing a full blood count was

**Table 1. Health facilities included in health facility assessments.**

|  | Clients per year | Number of beds |
|---|---|---|
| *Ghana* | | |
| *District 1* | | |
| Health facility 1 | 72,000 | 79 |
| Health facility 2 | 156,000 | 126 |
| Health facility 3 | 48,115 | 60 |
| Health facility 4 | 9451 | 16 |
| *District 2* | | |
| Health facility 1 | 40,511 | 64 |
| Health facility 2 | 30,000 | 100 |
| Health facility 3 | 42,580 | 70 |
| Health facility 4 | 36,000 | 18 |
| *District 3 (pilot district)* | | |
| Health facility 1 | 57,707 | 300 |
| *Uganda* | | |
| *District 1* | | |
| Health facility 1 | 19,000 | 100 |
| Health facility 2 | 500* | 68 |
| Health facility 3 | 51,100 | 400 |
| Pilot facility | 1200 | 57 |
| *District 2* | | |
| Health facility 1 | 30,000 | 100 |
| Health facility 2 | 24,000 | 120 |
| Health facility 3 | 37,200 | 300 |

* This was a private facility with fewer client numbers than other facilities.

**Table 2. Summary of qualitative data collection.**

|  | Number of participants |
| --- | --- |
| **Ghana** | |
| *District 1* | |
| Focus group 1 (blood donors) | 4 |
| Focus group 2 (community leaders) | 8 |
| Focus group 3 (pregnant or recently delivered women) | 7 |
| Focus group 4 (male partners of pregnant or recently delivered women) | 7 |
| In-depth interviews: health facility 1 | 3 |
| In-depth interviews: health facility 2 | 3 |
| In-depth interviews: health facility 3 | 2 |
| In-depth interviews: health facility 4 | 3 |
| Key informant interviews | 5 |
| **Total** | **42** |
| *District 2* | |
| Focus group 1 (blood donors) | 6 |
| Focus group 2 (community leaders) | 6 |
| Focus group 3 (pregnant or recently delivered women) | 10 |
| Focus group 4 (male partners of pregnant or recently delivered women) | 6 |
| In-depth interviews: health facility 1 | 3 |
| In-depth interviews: health facility 2 | 3 |
| In-depth interviews: health facility 3 | 3 |
| In-depth interviews: health facility 4 | 3 |
| Key informant interviews | 2 |
| **Total** | **42** |
| **Total (all Ghana participants)** | **84** |
| **Uganda** | |
| *District 1* | |
| Focus group 1 (blood donors) | 8 |
| Focus group 2 (community leaders and elders) | 7 |
| Focus group 3 (pregnant or recently delivered women) | 8 |
| In-depth interviews: health facility 1 | 2 |
| In-depth interviews: health facility 2 | 3 |
| In-depth interviews: health facility 3 | 4 |
| Key informant interviews | 5 |
| **Total** | **37** |
| *District 2* | |
| Focus group 1 (blood donors) | 8 |
| Focus group 4 (partners of pregnant or recently delivered women) | 6 |
| In-depth interviews: health facility 1 | 3 |
| In-depth interviews: health facility 2 | 3 |
| In-depth interviews: health facility 3 | 2 |
| Key informant interviews | 4 |
| **Total** | **26** |
| **Total (all Uganda participants)** | **63** |

Results from both the facility assessment and qualitative data collection are presented together under the following headings: diagnosing anaemia in pregnancy; localised understanding of anaemia in pregnancy; and ANC and the prevention and management of anaemia in pregnancy. Subheadings under each category reflect emergent themes from qualitative data.

**Table 3. Maternal anaemia equipment and process availability.**

| | Ghana | | | Uganda | |
|---|---|---|---|---|---|
| Question | District 1 | District 2 | District 3 (pilot) | District 1 | District 2 |
| Are there standards and protocols for anaemia identification at this facility? | 3/4 Not available 1/4 Reported available, not seen | 3/4 Not available 1/4 Unsure/don't know | 1/1 Not available | 2/3 Not available 1/3 Reported available, not seen | 3/4 Not available 1/4 Reported available, not seen |
| Which haemoglobin tests are available at this facility?* | 4/4 Automated analyzer 1/4 Spectrophotometer | 4/4 Automated analyzer 1/4 Manual test 1/4 Haemoglobin meter | 1/1 Automated analyzer | 3/3 Automated analyzer 1/3 DiaSpect 1/3 Calorimeter 1/3 Hematocrit centrifuge | 3/4 Automated analyzer 1/4 Hemocue 1/4 Manual test 1/4 Hemoglobinometer |
| Are there point of care Hb tests available at this facility | 1/4 Unsure/don't know 3/4 No | 1/4 Yes 3/4 No | 1/1 Unsure/ don't know | 3/3 Not available | 3/3 Not available ** |
| Does this hospital regularly give iron and folic acid to pregnant women? | 4/4 Yes to all pregnant women | 4/4 Yes to all pregnant women | 1/1 Yes to all pregnant women | 3/3 Yes to all pregnant women | 4/4 Yes to all pregnant women |
| Does this hospital regularly give intermittent malaria prophylaxis to pregnant women? | 4/4 Yes to all pregnant women | 4/4 Yes to all pregnant women | 1/1 Yes to all pregnant women | 3/3 Yes to all pregnant women | 4/4 Yes to all pregnant women |
| Does this hospital routinely give deworming tablets to pregnant women? | 1/4 Only prescribed to women living in higher-risk areas 3/4 Yes to all pregnant women | 3/4 Only prescribed to women living in higher-risk areas 1/4 Yes to all pregnant women | 1/1 Yes to all pregnant women | 3/3 Yes to all pregnant women | 4/4 Yes to all pregnant women |
| Proportion of women presenting in labour that had their Hb measured at the time of labour in the past three months | 12/354*** (3%) | Data not available | 1029/1029 (100%) | 27/56**** (48%) | 208/1105 (22%) |

*All available tests were mentioned, which was typically more than one per facility.

**This question was not asked in the pilot district in Uganda, which is within district 2.

*** Note the totals here exclude one facility (with 455 women coming in labour) as anaemia measures at the time of labour were not routinely documented.

****Note the totals here exclude one large facility (with 1490 women coming in labour) as anaemia measures at the time of labour were not routinely documented.

the most common method by far, being stated by participants across all facilities in the assessment in Ghana and in 6/7 (86%) facilities in Uganda. Only two facilities each in Ghana and Uganda also used non-automated methods (e.g. colorimeter, haematocrit) for measuring or estimating haemoglobin concentration. In Uganda, only one facility reported the use of Hemocue and another reported the use of DiaSpect, both of which can be point-of-care tests. However, no facilities in either country reported having haemoglobin measurement that was explicitly for point-of-care use (Table 3).

In the facility assessment, in Ghana, only 5/9 (56%) of facilities had available documentation on assessment of haemoglobin at the time of labour, and this was reportedly done for all women in two of those three facilities (1029/1029 women in one facility, and 6/6 women in another), for 6/10 women coming to the facility for childbirth in the third, and for zero women in the other two. In Uganda, these data were available in 6/7 (86%) facilities, but of the 1161 women presenting during labour in the three months prior to the assessment across these six facilities, only 235 (20%) of women had their haemoglobin assessed at the time of labour. However, most of these came from one facility, where 116/156 (74%) of women had haemoglobin assessed during labour—the range across facilities was 0–74%, with three facilities at roughly 10% and one at 59% of women (Table 3).

Maternity staff in Uganda noted that haemoglobin testing was done at various times and frequencies during pregnancy. Only one IDI participant in Uganda mentioned measuring

haemoglobin during labour. At the facilities in Ghana, most or all women had their haemoglobin measured during labour as routine practice, which was confirmed in IDIs.

*Yes it's a routine thing we do here, all [labour] admissions that come, they go through the lab, do full blood count, then we know your haemoglobin level before any other thing.* (IDI, matron in-charge, Ghana)

In community FGDs, especially those with women, in both countries, participants did note that their "blood is measured" during ANC.

It was also mentioned in both countries—though considerably more in Uganda—that, in the absence of haemoglobin testing, anaemia would be clinically assessed by pallor. Some district-level key informants in Uganda noted that lab-based haemoglobin testing is not always available, leading to this reliance on clinical assessment, which may not be as objective.

*Clinical assessment depends entirely on the clinical acumen of the individual and it is highly subjective, so we want to beef it up, we want to support the clinical assessment with the laboratory confirmation, you cannot depend on just clinical assessment.* (KII, district official, Uganda)

In Uganda, some facility-based participants also reported that a barrier to the laboratory diagnosis of anaemia in pregnancy was that sometimes error—human or technical—could lead to inaccuracies, for example, through documentation or equipment errors.

*Respondent: When they mix the results, they may end up giving low blood count to the patient who is actually okay. . . [Or] when the machine for measuring haemoglobin is not working very well, it can give a false haemoglobin and we end up transfusing blood. (IDI, midwife, Uganda)*

## Anaemia in pregnancy is taken seriously by healthcare providers

In both countries, it was recognised that a woman should not be anaemic at the time of birth. Iron supplementation and nutritional counselling were cited as key ways to prevent or manage anaemia, though it was recognised by many participants that more severe anaemia may require blood transfusion.

*When [haemoglobin] is 10 [g/L], they are being given haematinics for one month but when its nine and below, they are being referred to doctor to be given haematinics, mostly tot'hema [oral iron solution]. But if it's seven, seven and below, they are being counselled on transfusion, yes.* (IDI, matron in-charge, Ghana)

In both countries, maternity care providers commented on their commitment to following up with women they have identified as anaemic.

*Once we see [anaemia], during the antenatal, we make sure that you don't go*

*down. Once you come here and I see you, I monitor you seriously. At least every week or two; at-least two weeks or third week, we do the blood test again to see if it has come up. And we have been successful in those cases.* (IDI, matron in-charge, Ghana)

These participants noted difficulties in ensuring continuity of care, which acted as a barrier to effective anaemia in pregnancy management, often due to gaps in ANC attendance by pregnant women, including those with anaemia.

*. . .ones who just came with maybe anaemia in pregnancy, those are actually the hardest to come back, but these others to them they think there are justifiable causes, like maybe I ruptured my uterus or something, those ones will come back.* (IDI, matron in-charge, Uganda)

To encourage attendance at ANC, providers usually give women the details of their next ANC appointment at their current one so that women are aware of the importance of coming back and know when they should next attend. In Ghana, maternity staff and community members noted that women are provided with midwives' contact details. This practice enabled better organisation of appointments, shorter waiting times, and provided a communication channel to address women's needs, particularly for those who struggle to access care.

*They also tell her days on which she should keep coming back to the hospital to check the status of her pregnancy until she gives birth.* (FGD, community influencers, Uganda)

*The staff, most of them give their numbers out, and with a good relationship, a client will always call.* (KII, district official, Ghana)

### Localised understanding of anaemia in pregnancy

**Anaemia is understood according to its symptoms.** In Ghana, some community FGD participants had a good understanding of the symptoms of anaemia. In Uganda, the term anaemia was recognised but was again understood in terms of its symptoms. These included being pale, having low energy, and dizziness. Anaemia was generally understood to mean a "low amount of blood" or "low blood volume" and was typically thought to be caused by not eating and sleeping well. In Uganda, community FGD participants also regularly talked about swelling, and in the FGD with pregnant or recently delivered women, anaemia was also linked to weight loss and loss of appetite.

*Respondent 4: I know anaemia is when you don't have enough blood, or your blood level is low.*

Respondent 3: Please I don't know.

Respondent 2: I haven't heard of anaemia before.

Respondent 1: I haven't also heard it before.

Respondent 5: Same.

Respondent 6: Same.

Interviewer: What are some of the symptoms of anaemia?

Respondent 6: Sometimes you feel dizzy and not feel okay.

Respondent 2: Your heart beats faster and you easily get tired.

(FGD, pregnant or recently delivered women, Ghana)

Anaemia was seen as dangerous by participants in all community FGDs, especially to the developing fetus, leading to foetal death or premature childbirth. Some participants also noted

it was potentially life-threatening to the mother, though this was given less focus. In one FGD in Uganda, a male partner specifically noted anaemia means "less oxygen and nutrients for the baby."

Most FGD participants were aware of haemoglobin testing as a way to detect "blood volume"—i.e. anaemia. Male partners in Ghana and community influencers in Uganda also noted that there is a normal haemoglobin range or a "cut-off" and that falling below these indicates anaemia.

Reflecting on what was noted by facility- and district-level participants, some women in Uganda also described how a clinical assessment for anaemia is done during ANC.

*The doctors instructs you to look up and looks into your eyes and looks into your hands to see if you have blood.* (FGD, pregnant or recently delivered women, Uganda)

**There is understanding of local foods associated with "improving blood".**   In Ghana, participants across community FGDs could describe many local foods that were useful in combatting anaemia to "help their blood", including leafy greens, legumes, plantains, cabbage, milk, dried fish, meat, and eggs. Tomato paste and coke were also mentioned. Community influencers in Ghana also made a link between anaemia and vitamin C deficiency, noting the importance of fruit for preventing anaemia. Pregnant or recently delivered women also mentioned multivitamins and "iron 3 [a hematinic oral solution]—it's a blood tonic" (FGD, pregnant women, Ghana).

Key informants in Ghana described several successful programmes to educate pregnant women and community members about anaemia in pregnancy. These include pregnancy schools—which are educational sessions run outside of routine antenatal appointments—and dietary education provided through two community-based nutrition programmes.

*. . . we used to have anaemia quite bad in pregnant women, so we put in two major programs. Food demonstration and food bazaars.* (KII, district official, Ghana)

In Uganda, local foods widely understood to prevent anaemia or to be consumed by women with anaemia in pregnancy are fruits (mango, banana, watermelon, pawpaw, pineapple, avocado), vegetables (eggplant, amaranth leaves and other greens), maize meal porridge, ground nuts, beans, and meat.

## ANC and the prevention and management of anaemia in pregnancy

**Perceptions towards ANC are very positive.**   Across all community FGD participants, ANC was seen as important.

*There are many diseases that come up in the pregnancy period. . .the baby can even die at the point of birth if these things are not taken care of, so it is necessary to go*

*for ANC.* (FGD, male partners, Ghana)

*We are firm in knowing we won't meet any complications because we have been attending ANC.* (FGD, male partners, Uganda)

Across community FGDs in Ghana, services referred to as part of ANC included: checking the baby's growth and position; checking the mother's weight, blood pressure, and temperature; as well as doing tests for infections and providing treatment where necessary. The

provision of counselling, especially around nutrition and hygiene, was also noted frequently. A few participants in Ghana also mentioned that screening for Rhesus factor and sickle cell disease would be done. Male partners in Ghana were divided into those who were unsure about what ANC entails and those who had more knowledge as they had participated in ANC or asked their wives about it and reviewed ANC booklets. Only community influencers suggested that couples receive preferential treatment if a male partner accompanies a woman to ANC. However, mandatory HIV testing during ANC was mentioned as a deterrent to male partner involvement.

> *I have no idea. . .no, because I am not allowed inside.* (FGD, male partners, Ghana)

> *My wife said they tested her [haemoglobin] and was told her blood level is low, so I called the nurse and asked her because I didn't see it in the antenatal book, so the nurse confirmed and said yes, she was told, but they didn't write it in the book.* (FGD, male partners, Uganda)

Community participants in Uganda were quite specific about what ANC covers, including measuring the "quantity of your blood" (to test for anaemia), screening for infections like HIV and syphilis and treating these, providing an expected due date, checking for blood pressure and blood sugar, counselling on nutrition and hygiene, the position and health of the baby, the amount of amniotic fluid, counselling on the place of childbirth, and education on understanding what different pains in pregnancy mean. Some also mentioned "mama kits" that include labour and childbirth supplies as well as newborn necessities that are given depending on the health facility they visited—it was expressed that they would like to see this given at all public facilities.

**ANC timing and frequency are not always understood.** Guidelines in both contexts indicate that there should be eight ANC visits throughout pregnancy and that the first visit should be initiated within the first trimester to ensure that interventions are offered at the appropriate time. FGDs with pregnant or recently delivered women in Ghana highlighted variable knowledge about the timing and number of ANC visits, with only one participant noting that there should be "seven or eight" ANC visits throughout pregnancy. Timing of ANC initiation varied from two weeks to six months, though of the 23 pregnant women participating across three FGDs, 16/23 (70%) noted starting ANC in the first trimester. Women who sought care very early (within the first or second month of pregnancy) often commented that they had felt particularly unwell or that they had a history of miscarriage, which prompted them to seek care.

This variability in knowledge about ANC visits was also reflected in male partner FGDs in Ghana, but there seemed to be a stronger sense that going early was not required if there were no issues and that most women went after six months. However, when asked when women *should* go for ANC, there was consensus that starting at around three months, though a handful of participants noted that ANC attendance should begin as soon as the pregnancy is identified. When asked how often ANC occurs, there was a wide range of responses, from every two or three weeks to monthly, for a total number of visits ranging from 6–18 over the course of pregnancy. There was the sense amongst participants that ANC attendance is increasing, primarily due to policies supporting free care.

In FGDs with community influencers in Ghana, there was broad agreement around the start of ANC being from two-to-three months up until the pregnancy is visible. They expressed that women might delay going so that the pregnancy is more advanced, so they only have to pay for one ultrasound.

> *Interviewer: How many times [should a pregnant woman attend ANC] . . .? Number 3.*

*Respondent 3: 3 times.*

*Interviewer: 3 times.*

*Number 4 how many times . . .?*

*Respondent 4: It can be 6 times.*

*Interviewer: 6 times? Number 6 how many times?*

*Respondent 6: 9 times.* (FGD, community influencers, Ghana)

Across all community FGD participants in Ghana, there was the perception that strong women and those with healthy pregnancies do not need ANC, and those who are "weaker" or with obvious health issues are the ones who need ANC earlier.

*It depends on the individual. Some people are physically strong when they are pregnant. Others aren't that strong, so they visit the hospital in their first week.* (FGD, community influencers, Ghana)

In Uganda, most participants in the FGD with pregnant or recently delivered women spoke about the timing of the first ANC visit as 1–3 months, and it was clear that "eight ANC visits" was correctly understood as being the desired number of visits. However, in the FGD with male partners and both FGDs with community influencers, this ranged from four-to-eight. There was a shared understanding that most women would be likely to seek out ANC as soon as they have a positive pregnancy test but that there would be women who might go later in the absence of complications.

*They go for antenatal on the sixth month of pregnancy. . .because until then there have not been any complications, so there would not be a need to go for antenatal.* (FGD, male partners, Uganda)

However, community influencers suggested that most women start care from three-to-five months' gestation and associated women coming later with a lack of responsibility.

*[Coming at five months for the first ANC] is for a mother that has delayed or. . .doesn't care. . .at three months, a responsible mother should be coming from ANC.* (FGD, community influencers, Ghana)

*Doctors taught them [women in communities] that when you get pregnant, you start coming for ANC. (FGD, community influencers, Uganda)*

**Persistent barriers to ANC uptake exist.** All community FGD participants in both countries described recurring barriers to ANC care. These largely centred distance to the health facility, difficulty finding and paying for transportation—especially in remote areas—and other financial barriers, including "hidden costs" for equipment or medication that are expected to be free of charge. These barriers are exacerbated when repeated visits are required.

Some pregnant or recently delivered women in Ghana—especially those who already have children—suggested that expenses are increasing. There is a small charge for each visit (10 cedis (~1.7 USD at the time of data collection)) and two cedis (~0.34 USD) for some medications, even with health insurance. Male partners in Ghana agreed that there are many costs

associated with ANC and pregnancy, mainly linked to drugs and transportation, and estimated total expenses throughout pregnancy and childbirth as high as 1500 cedis (~255 USD). Although Ghana's National Health Insurance should cover costs, community-based participants regularly cited cost-related barriers. For example, if medication is required, as they explained, there are hidden costs that can influence the decision to seek care. In Uganda, it was widely agreed across community participants that ANC in public facilities is free, though payment is expected in private facilities. However, costs may still be incurred, as women might be sent to another facility if certain equipment is unavailable (e.g. an ultrasound machine), or they may be asked to buy medications.

*They also like to take money too much, whether you have insurance or not. Medicine that will cost 20 cedis you end up spending 100 cedis.* (FGD, community influencers, Ghana)

Respondent 1: It comes down to a money issue and fatigue.

*Respondent 3*: *I want to buttress that point. If you go today to do laboratory work, you would be expected to pay. After three days' time, when you are scheduled to come back, you would be asked to go and do another. So if that continues, then you find out that your money is getting finished. . .if you have money, then you go and give birth. They say it's free, but it's actually not.* (FGD, pregnant or recently delivered women, Ghana).

Many participants in both countries reiterated that the cost of ANC sometimes results in delays in women seeking and accessing anaemia in pregnancy diagnosis and management.

*When they go early they will do so many scans, so if you go late in the last trimester that means you will do like just one scan and you leave. So, I think it's the financial obligation that makes them stay longer at home.* (FGD, community influencers, Ghana)

Poor attitudes of ANC staff were extensively highlighted in both countries by pregnant women and male partners, as well as long wait times in Ghana. These prevented women from taking up ANC and also from giving birth in a health facility. Community influencers also noted that healthcare providers do not consistently provide good quality care and can be unhelpful and unkind. However, some women stated they were satisfied with the care provided and the relationship with their midwives. It was understood that this may affect the reputation of the health facility in positive or negative ways as women share information about their experiences.

*If I brought my wife here and if she wasn't treated well—or even if she was treated well—now they are advising [other women]. . .which will cause a change of heart on the other person.* (FGD, male partners, Uganda)

Local perceptions of having safely had children previously without seeking ANC or skilled attendance at birth may reinforce this idea among those women or women close to them.

*Some people have given birth safely without antenatal, so they don't see the need.* (FGD, pregnant or recently delivered women, Ghana)

In Uganda, the perception of having lower socioeconomic status was seen as a barrier to some women. Lack of permission or support from husbands or mothers-in-law was also repeatedly cited across community participants.

*Lack of proper clothes to wear to come to the hospital. . .they fear being seen in torn clothes and decided to keep home.* (FGD, pregnant or recently delivered women, Uganda)

The use of traditional care was frequently mentioned by community-based participants in Ghana. Traditional care was appreciated as it lacked many of the barriers of formal ANC, such as cost and distance. Traditional medicine was also sometimes viewed as more effective.

*Respondent: It is because sometimes traditional/herbal medicine is more potent than orthodox medicine. . .that is also a contributing reason.* (FGD, pregnant or recently delivered women, Ghana)

Facility-based participants and key informants explained that inadequate staffing is sometimes a barrier to adequate ANC and, therefore, to anaemia in pregnancy diagnosis and management, as laboratory personnel were in particularly short supply.

*Interviewer: So currently staff is a challenge?*

*Respondent: It's a huge problem, it's a huge problem.* (KII, district official, Ghana)

Staff training on anaemia in pregnancy appeared to be infrequent in both countries, though there was training before qualifying and on-the-job. Facility-based participants communicated gaps in training, particularly for training on attitude and conduct.

*Apart from training on certain topics, ours have been informal discussions when we go on rounds, we talk a lot and try to educate them on transfusion, that has been my approach, I don't normally run a lot of formal workshops and those things, we see a patient we just discuss it.* (IDI, midwife, Ghana)

Some district-level key informants in Uganda also noted the importance of continuing medical education for anaemia in pregnancy.

**Preventing and managing anaemia are well-understood by healthcare providers and less well understood by community members.**   In IDIs with maternity staff in both countries, alongside haemoglobin measurement, routine practices during ANC—including preventing and treating malaria, providing deworming medication, giving iron and folate supplements, and providing nutritional counselling—were described as important for preventing and managing anaemia in pregnancy. The use of insecticide-treated bednets was also mentioned by maternity staff and community-based participants. In Ghana, maternity staff reported that women are supervised in taking malaria medication to increase adherence. Women across FGDs in both countries noted that many women did not always like taking antimalarials or iron and folate supplements due to side effects.

*Yeah, we are doing it actually in antenatal, we start from antenatal, we first do haemoglobin, then on subsequent visits, we do it, and then that's why we give them ferrous and folic, whatever we have in place, even the minister of health, that's what she is advocating for, even the [intermittent preventative treatment for malaria] we are giving, if the woman does not have malaria, if she is dewormed and worms are not sucking blood, if she is given these to supplement her blood, then there is nothing that should stop her from fighting anaemia. Then also advising her on what to eat, nutritional status.* (IDI, matron in-charge, Uganda)

*I want to find out why they give us the malaria drug. When I went, they ask us to take the medicine right there. But when I came home I had adverse reactions; my mouth area was itching and when I touched it, rashes developed.* (FGD, pregnant or recently delivered women, Ghana)

These findings are echoed in facility assessments, where intermittent malaria prophylaxis is routinely given to all women across all participating facilities in both countries and iron and folate supplements are provided to all women during ANC. In Ghana, the provision of anti-helminth medications was offered to all women in ANC in 5/9 (56%) facilities and only to women in higher-risk areas in 4/9 (44%) facilities. In Uganda, these medications were offered to all women attending ANC (Table 3).

Participants in the FGDs with pregnant and recently delivered women in Ghana linked iron and folate supplements to the development of the baby's blood and bones rather than with their own anaemia. "I think folic acid supports blood cells. . .about iron? I have no idea" (FGD, male partners, Ghana). In Uganda, pregnant or recently delivered women knew pills could be given for anaemia but did not refer to these by name.

*They give us those red tablets to increase our blood levels. . .and. . .add on our blood levels and [ensure we] have enough blood to take care of the baby in the womb.* (FGD pregnant women, Uganda)

Iron and folate supplements were known by a few male partners and community influencers in Uganda to help women "gain blood". However, it was mostly understood as being important for health in general and not linked to resolving anaemia.

Although malaria and helminth infections were understood as bad for the baby and mother, they were not associated with anaemia in FGDs with pregnant or recently delivered women in Ghana. Only one pregnant or recently delivered participant in Uganda noted, "deworming prevents [pregnant women] from being anaemic". However, some male partners and community leaders in Ghana did note the association between malaria and "reducing blood", and one community influencer in Uganda noted malaria causes "blood loss and complications" and was therefore linked to anaemia. Only one male partner in Uganda noted that worms "take nutrients", resulting in the mother becoming malnourished and then anaemic (Male partner FGD, Uganda).

Blood transfusion was sparingly mentioned in both countries by community participants as a treatment for severe anaemia.

## Discussion

This mixed methods assessment explored facility readiness and diverse stakeholder perspectives to identify many good practices in Ghana and Uganda around the diagnosis, prevention, and management of anaemia in pregnancy, as well as several gaps.

Standard operating procedures or guidelines specific to anaemia in pregnancy were reported as sparingly present in both countries but were never seen. Ensuring that these are available, widely publicised, and used for continuing medical education would help to promote better practice and reduce anaemia in pregnancy and its consequences for the mother and baby. Such guidelines could be adapted from the "World Health Organization recommendations on ANC for a positive pregnancy experience", which recommend daily (or intermittent if side effects are intolerable) oral iron and folic acid supplementation and for pregnant women to be advised about food sources of vitamins and minerals, and dietary diversity [5].

Community participants widely understood the use of haemoglobin testing for detecting anaemia. Haemoglobin measurement around the time of labour did not appear to be routine in many facilities in Uganda, but is very important so that anaemia can be corrected prior to delivery, especially where women may have had inconsistent attendance of ANC and do not have a recent measure [17]. Doing so may reduce the occurrence and severity of peripartum haemorrhage [18]. In both countries, automated analysers were by far the most common way to measure haemoglobin; this is the recommended method for diagnosing anaemia in pregnancy [5]. However, the limited availability of laboratory personnel and equipment sometimes necessitated the use of manual haemoglobin measurements or clinical assessment of anaemia, even though clinical examination is inadequately sensitive and specific for diagnosing anaemia [19]. Point-of-care haemoglobin tests may be useful in this situation as they can be performed by non-laboratory personnel and can yield timely, reliable results for quick decision-making [20,21]. However high volume use of these tests can be expensive [20,22], limiting their availability in many contexts, as was also reflected in our assessment.

Overall, there was a good level of understanding of some critical aspects of anaemia. Community participants in both countries recognised the symptoms and consequences of anaemia but knew less about the causes. Its implications on the developing foetus, and to a lesser extent on the mother, were acknowledged by community participants. There was wide general knowledge about local foods that may help in preventing or reducing anaemia across both countries, including an emphasis on vitamin C-rich foods, which can increase the absorption of iron, especially from plant foods [23,24]. Dietary factors can play an important role in reducing the risk of anaemia [25]. Positively, it was clearly noted by participants in both contexts that nutrition education for anaemia prevention or mitigation is a common aspect of ANC.

Community participants unanimously understood ANC to be very important. The timing of uptake and frequency of ANC visits were less well-understood, especially among Ghanaian participants. In both countries, initiation of ANC was frequently (and correctly) cited by participants as being within the first three months of pregnancy. However, literature from both Uganda and Ghana suggest this is not typical of most women, for whom early initiation of ANC and regular attendance is a persistent challenge due to gaps in knowledge and social, financial, and geographical barriers to accessing care [26–29]; achieving all eight ANC contacts occurs for a minority of women [30–32]. Community education about the importance of early and consistent ANC uptake may, therefore, be beneficial in managing anaemia in pregnancy [33,34].

Anaemia in pregnancy was taken seriously by maternity staff, with efforts being made to follow-up with patients identified as anaemic, including providing personal phone numbers and appointment reminders. However, despite efforts made, this follow-up was sometimes difficult due to women's reluctance to take up care. In both countries, there were common barriers around ANC uptake, largely stemming from long distances to the health facilities, difficulty finding or paying for transportation, other costs, especially for medications, and poor staff attitudes. From the health providers' perspectives, staff shortages and a lack of anaemia in pregnancy-specific training were noted in both countries. All of these resonate with findings from other lower-resource settings [35,36]. Without early and consistent uptake of ANC, women miss opportunities for interventions that can prevent or mitigate anaemia in pregnancy, potentially leading to poorer health outcomes for the mother and the baby [37]. It is, therefore, important that initiatives to reduce barriers to accessing ANC are implemented. For example, in other African settings, women have been provided with transportation vouchers that they can use in exchange for bus or motorcycle transport, which helps to overcome some of the transportation barriers related to attending ANC [38–40].

Finally, though the provision of antimalarials, iron and folate supplements, and deworming are routine during ANC in both countries, very few community participants understood their link with preventing or managing anaemia in pregnancy. Even where the link was made, it was not fully understood—for instance, helminth infections can cause anaemia due to red cell destruction or intestinal bleeding [41], but the women associated them with causing malnutrition. Iron and folate supplements are very effective in reducing anaemia if taken consistently during pregnancy [42,43] and ANC clinic consultations are a critical point through which iron and folate supplements are provided. Delays in care may, therefore, exacerbate the risk of developing anaemia [44]. Further, across many contexts, knowledge about iron and folate supplements and their importance supported significant improvements in adherence throughout pregnancy [45–47]. In a recent systematic review of iron and folate supplement adherence, the most significant contributor to compliance was knowledge of anaemia and the role of iron and folate supplements in preventing or managing it [47]. As participants highlighted the importance of counselling on various topics in ANC and noted a clear understanding of anaemia as dangerous for the developing foetus and mother, emphasising the link between iron and folate supplements, antimalarials, deworming, and anaemia within this counselling is an easy-to-action and likely highly effective step in promoting adherence to these treatments given during ANC.

## Strengths and limitations

This study benefitted from having both quantitative and qualitative components, drawing insights across a wide range of respondents. Collecting data from two countries enabled comparison around the many similarities and few differences. Our study focused on surgery- and blood transfusion-capable facilities, so the findings may not apply to lower-level facilities, which may be less well-resourced and have less experienced or knowledgeable staff. Further, as we only collected data from 2–3 districts (and 16 facilities total) across both countries, our findings may not be generalizable. However, the many similarities across both countries suggest that our findings may be applicable to other facilities providing comprehensive obstetric care across Sub-Saharan Africa.

## Conclusions

Limited capacity for haemoglobin testing, difficulties in accessing ANC, and staffing issues were raised by participants in both countries. Improved education for pregnant women, their families, and communities on anaemia, including its importance, causes, treatment and prevention, may support better uptake of anaemia prevention and treatment strategies. Under-staffing, insufficient resources, and provision of and training on standards of care for anaemia in pregnancy should be addressed by the health service at the district or national levels.

## Supporting information

**S1 Checklist. Checklist—contains our human subject research checklist.**
(PDF)

**S1 Data. Data collection instruments—contains our data collection instruments.**
(PDF)

## Acknowledgments

We thank all of the participating health facilities, staff, and community members for their insights and cooperation.

## Author Contributions

**Conceptualization:** Tara Tancred, Vincent Mubangizi, Imelda Bates, Bernard Natukunda, Lucy Asamoah Akuoko.

**Data curation:** Emmanuel Nene Dei, Syliva Natukunda, Daniel Nana Yaw Abankwah.

**Formal analysis:** Tara Tancred, Phoebe Ellis.

**Funding acquisition:** Tara Tancred, Imelda Bates, Bernard Natukunda, Lucy Asamoah Akuoko.

**Investigation:** Tara Tancred, Syliva Natukunda.

**Methodology:** Tara Tancred, Vincent Mubangizi, Emmanuel Nene Dei, Syliva Natukunda, Daniel Nana Yaw Abankwah, Imelda Bates, Lucy Asamoah Akuoko.

**Project administration:** Tara Tancred, Bernard Natukunda.

**Supervision:** Tara Tancred, Vincent Mubangizi, Daniel Nana Yaw Abankwah, Imelda Bates, Bernard Natukunda, Lucy Asamoah Akuoko.

**Validation:** Tara Tancred, Syliva Natukunda, Daniel Nana Yaw Abankwah.

**Writing – original draft:** Tara Tancred.

**Writing – review & editing:** Tara Tancred, Vincent Mubangizi, Emmanuel Nene Dei, Syliva Natukunda, Daniel Nana Yaw Abankwah, Phoebe Ellis, Imelda Bates, Bernard Natukunda, Lucy Asamoah Akuoko.

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
