## [Decision Letter · Decision Letter 0]

27 May 2024

PGPH-D-24-00750

Prevention and management of anaemia in pregnancy: community perceptions and facility readiness in Ghana and Uganda

Dear Dr. Tancred,

Thank you for submitting your manuscript to PLOS Global Public Health. After careful consideration, we feel that it has merit but does not fully meet PLOS Global Public Health’s publication criteria as it currently stands. Therefore, we invite you to submit a revised version of the manuscript that addresses the points raised during the review process.

Editor comments:

The reviewers and I found the manuscript to be well-written, and offers helpful cross-country insights into community perceptions and facility capabilities concerning the identification and management of anemia during pregnancy. The inclusion of male partners is particularly interesting. However, the reviewers and I also have several questions.Concerning the methods, please see the comments from the reviewers requesting additional information. I would also encourage the authors to review COREQ or similar reporting guidelines to add additional information concerning the qualitative methods. The quantitative methods could also use a little more description, in terms of why the sample size was chosen, and why further analyses were not conducted with the data. I also encourage the authors to publish their data collection instruments and final qualitative codebook as appendices/supplemental materials. The choice of Ghana and Uganda for the work needs to be explained--I understand that this was a sub-study, but it still isn't clear why these two countries were chosen. Finally, the authors need to explain how this is a mixed methods study. The type of mixed methods design should be named, and the point(s) of integration of qual and quant methods/data needs to be explained. As is, this seems more like a multi-method study, without integration between quant and qual findings, rather than a mixed methods study.Consider expanding the limitations concerning the quantitative data, if a power/sample size analysis was not the basis for the quantitative data collection, which may in turn have limited the analyses possible.

We look forward to receiving your revised manuscript.

Kind regards,

Marie A. Brault, PhD

Academic Editor

Journal Requirements:

Additional Editor Comments (if provided):

Reviewers' comments:

Reviewer's Responses to Questions

**Comments to the Author**

1. Does this manuscript meet PLOS Global Public Health’s publication criteria? Is the manuscript technically sound, and do the data support the conclusions? The manuscript must describe methodologically and ethically rigorous research with conclusions that are appropriately drawn based on the data presented.

Reviewer #1: Yes

Reviewer #2: Yes

2. Has the statistical analysis been performed appropriately and rigorously?

Reviewer #1: N/A

Reviewer #2: Yes

3. Have the authors made all data underlying the findings in their manuscript fully available (please refer to the Data Availability Statement at the start of the manuscript PDF file)?

Reviewer #1: Yes

Reviewer #2: No

4. Is the manuscript presented in an intelligible fashion and written in standard English?

Reviewer #1: Yes

Reviewer #2: No

5. Review Comments to the Author

Reviewer #1: Comments to Authors

The manuscript is well-written and addresses an important issue prevalent in low-resource settings. It also contributes to the body of literature on the subject.

Kindly address the following:

1. The citations should be inserted before punctuation marks.

2. The methods section should include more details on the sample size for the FDGs, IDIs, and KIIs.

3. Line 119: The Methods section should have a heading for “quantitative data” and “qualitative data.”

4. Lines 128-132: Please provide a citation for this methodology if available.

5. Line 158: how were the IDIs, FGS, and KIIS interviews recorded? Written or audio recording? Also consider adding the sample sizes for each of the IDIs, FGS, and KIIS. E.g IDIs (n=x)

6. Line 167: What are these tools and what do they measure?

7. This statement is confusing. “Since none of the tools needed minimal revision following piloting, we included data from the pilots in our analysis.” Did you mean the tools needed minimal revision?

8. Line 181: Consider adding the word “Quantitative” to the phrase “data from the health facility assessments..”

9. Table 1: Add a row for “Total” for each district and Country separately.

10. Table 2: consider adding demographic information if available (Gender and Age)

11. The results section should also include a table for the qualitative findings on the “availability of appropriate standards or protocols, drugs, equipment, trained staff, and infrastructure to support anaemia prevention and management”, “measurement of haemoglobin at the time of labour” as stated in the methodology. This should include the country, data sources and findings.

12. Consider: The result section should separate qualitative findings from quantitative findings. The findings should be reported under what the tools were measuring (especially for the quantitative findings).

13. Line 267: the statement should be restructured. Human error and equipment error are two different things.

14. Lines 313-315: Is this a statement from multiple sources? I see (FGD, community influencers, Uganda). Please clarify- This format is also repeated many times in the manuscript.

15. Lines 349-359 appear to fit under different topics (consider a new heading for it)

16. Line 370: What type of program? Food and nutrition education programs?

17. Line 423: Should this be italicized? It appears to be a subheading under “ANC and the prevention and management of anaemia in pregnancy”

18. Lines 427-429: What was the average number or range of ANC visits mentioned by respondents?

19. Lines 531-532 are confusing. Please restructure.

20. Lines 534-537: Is this by women in the FGDs or the Community influencers? The succeeding quotation in lines 539 to 541 is by multiple sources (refer to comment 15 above).

21. Lines 611-616: this should also be stated earlier under the results from quantitative findings

22. Line 645: consider restructuring to “This mixed methods assessment explored facility readiness and diverse stakeholder…”

23. Lines 679-681: Add some sentences on the value of this for reducing anaemia among pregnant women. This knowledge can be further emphasized during ANC.

24. The discussion should also discuss financial barriers to accessing care during ANC as cited by the respondents.

25. Line 729: Limitations should include geographical limitations of the study sites especially for generalization purposes. The findings would only apply to similar facilities and populations in sub-Saharan Africa.

Reviewer #2: Thank you for sharing your work with us. It was an eye-opening experience reading your work. However, there are several points I would like to share.

1.First, related to the references, some of your references are outdated (older than 5 years) and irrelevant to the sentence prior to the citation. When you put a

citation behind a sentence, the citation must justify the sentence.

2.Second, The use of the journal's template is advisable to improve the flow and readability of the manuscript. Follow the template as closely as possible.

3.Ensure that the presentation of tables comply with the journal's style and formatting guidelines.

4.Discussion of research results: The discussion should provide arguments for the research results that have been claimed, have a logical cause-and-effect explanation and be structured in the form of a 'new story'.

5.References are advisably from reputable journal articles.

6.Local references should be kept to a maximum of 10% from total references. Many of the references are not recent.

7.There enough is not detail in order to replicate the study.

8.The article needs to be a review of grammatical errors.

6. PLOS authors have the option to publish the peer review history of their article (what does this mean?). If published, this will include your full peer review and any attached files.

**Do you want your identity to be public for this peer review?** For information about this choice, including consent withdrawal, please see our Privacy Policy.

Reviewer #1: No

Reviewer #2: No

---

## [Decision Letter · Decision Letter 1]

1 Aug 2024

Prevention and management of anaemia in pregnancy: community perceptions and facility readiness in Ghana and Uganda

PGPH-D-24-00750R1

Dear Dr. Tancred,

We are pleased to inform you that your manuscript 'Prevention and management of anaemia in pregnancy: community perceptions and facility readiness in Ghana and Uganda' has been provisionally accepted for publication in PLOS Global Public Health.

Best regards,

Marie A. Brault, PhD

Academic Editor

Reviewer Comments (if any, and for reference):

Reviewer's Responses to Questions

**Comments to the Author**

1. If the authors have adequately addressed your comments raised in a previous round of review and you feel that this manuscript is now acceptable for publication, you may indicate that here to bypass the “Comments to the Author” section, enter your conflict of interest statement in the “Confidential to Editor” section, and submit your "Accept" recommendation.

Reviewer #1: All comments have been addressed

Reviewer #2: All comments have been addressed

2. Does this manuscript meet PLOS Global Public Health’s publication criteria? Is the manuscript technically sound, and do the data support the conclusions? The manuscript must describe methodologically and ethically rigorous research with conclusions that are appropriately drawn based on the data presented.

Reviewer #1: Yes

Reviewer #2: Yes

3. Has the statistical analysis been performed appropriately and rigorously?

Reviewer #1: N/A

Reviewer #2: Yes

4. Have the authors made all data underlying the findings in their manuscript fully available (please refer to the Data Availability Statement at the start of the manuscript PDF file)?

Reviewer #1: Yes

Reviewer #2: Yes

5. Is the manuscript presented in an intelligible fashion and written in standard English?

Reviewer #1: Yes

Reviewer #2: Yes

6. Review Comments to the Author

Reviewer #1: Authors appear to have addressed all my comments and recommendations.

Reviewer #2: Has this manuscript never been published in another journal?

7. PLOS authors have the option to publish the peer review history of their article (what does this mean?). If published, this will include your full peer review and any attached files.

**Do you want your identity to be public for this peer review?** For information about this choice, including consent withdrawal, please see our Privacy Policy.

Reviewer #1: No

Reviewer #2: No
